# Differential Effects of Aripiprazole on Electroencephalography-Recorded Gamma-Band Auditory Steady-State Response, Spontaneous Gamma Oscillations and Behavior in a Schizophrenia Rat Model

**DOI:** 10.3390/ijms25021035

**Published:** 2024-01-14

**Authors:** Florian W. Adraoui, Kenza Hettak, Geoffrey Viardot, Magali Alix, Sabrina Guiffard, Benoît Meot, Philippe L’Hostis, Anne Maurin, Eric Delpy, Christophe Drieu La Rochelle, Kevin Carvalho

**Affiliations:** 1Biotrial, Non-Clinical Pharmacology Department, 7-9 Rue Jean-Louis Bertrand, 35000 Rennes, France; florian.adraoui@biotrial.com (F.W.A.);; 2Biotrial, Neuroscience Department, 6 Avenue de Bruxelles, 68350 Brunstatt-Didenheim, France; 3Biotrial, Neuroscience Department, 7-9 Rue Jean-Louis Bertrand, 35000 Rennes, France

**Keywords:** schizophrenia, aripiprazole, gamma-band ASSR, spontaneous gamma oscillations, drug development, NMDAr hypofunction

## Abstract

The available antipsychotics for schizophrenia (SZ) only reduce positive symptoms and do not significantly modify SZ neurobiology. This has raised the question of the robustness and translational value of methods employed during drug development. Electroencephalography (EEG)-based measures like evoked and spontaneous gamma oscillations are considered robust translational biomarkers as they can be recorded in both patients and animal models to probe a key mechanism underlying all SZ symptoms: the excitation/inhibition imbalance mediated by N-methyl-D-aspartate receptor (NMDAr) hypofunction. Understanding the effects of commercialized atypical antipsychotics on such measures could therefore contribute to developing better therapies for SZ. Yet, the effects of such drugs on these EEG readouts are unknown. Here, we studied the effect of the atypical antipsychotic aripiprazole on the gamma-band auditory steady-state response (ASSR), spontaneous gamma oscillations and behavioral features in a SZ rat model induced by the NMDAr antagonist MK-801. Interestingly, we found that aripiprazole could not normalize MK-801-induced abnormalities in ASSR, spontaneous gamma oscillations or social interaction while it still improved MK-801-induced hyperactivity. Suggesting that aripiprazole is unable to normalize electrophysiological features underlying SZ symptoms, our results might explain aripiprazole’s inefficacy towards the social interaction deficit in our model but also its limited efficacy against social symptoms in patients.

## 1. Introduction

Schizophrenia (SZ) is a complex neuropsychiatric disorder characterized by clinical manifestations divided into three categories: positive (e.g., agitation, hallucinations and delusions), negative (e.g., social withdrawal and anhedonia) and cognitive symptoms (e.g., learning, memory and concentration issues as well as poor executive functioning) [1]. Unfortunately, there has been no major improvement in the therapeutic management of SZ since the first commercialization of the atypical antipsychotic clozapine in the 1970s. Atypical antipsychotics can effectively reduce positive symptoms, but their effects on the neurobiological underpinnings of SZ as well as cognitive and negative symptoms remain very limited [1,2,3]. 

Over the years, the lack of new innovative therapies targeting SZ neurobiology and cognitive and social symptoms has raised the question of the robustness and translational value of preclinical methods used during drug development [4]. Today, experts in the field consider quantitative electroencephalography (qEEG) and EEG-based event-related potentials (ERPs), such as the auditory steady-state response (ASSR), as potential powerful translational biomarkers for SZ drug development [4,5]. While qEEG allows measuring spontaneous neural oscillations in different frequency bands, the ASSR is a brain response evoked when auditory stimuli (trains of clicks or tones) produced at a given frequency are presented to a subject. The ASSR is then evoked by neuronal populations entrained at the same frequency and can be recorded using EEG in both human subjects and laboratory animals, providing translational information on evoked oscillations at specific frequencies and probing the integrity of neuronal pathways [4,5]. SZ patients display higher spontaneous gamma oscillations (30–80 Hz) and a reduced phase synchrony (i.e., the rate of neural response in phase with the stimuli, which is measured by the phase-locking factor (PLF) or the inter-trial coherence) of the ASSR when it is evoked at gamma frequencies (e.g., between 30 and 50 Hz) [6,7,8,9,10,11,12]. These abnormal electrophysiological features are present across various brain areas (e.g., in the frontal and auditory cortex) and thought to be consequences of N-methyl-D-aspartate receptor (NMDAr) signaling impairments (which makes the link with the SZ NMDAr hypofunction hypothesis [13]) and associated neuronal excitation/inhibition imbalance [6,8,10,11,12,14]. Importantly, these electrophysiological dysfunctions have been associated with all the symptoms of SZ and may thereby provide valuable translational information regarding the therapeutic potential of investigational drugs [15,16,17,18]. Used in combination with traditional behavioral methods, these biomarkers could foster the preclinical development of new medicines for SZ and allow further characterization of already commercialized antipsychotics to better understand their effects (or absence of effect), thus providing valuable insights for new drug development aimed towards SZ [19]. 

For SZ patients undergoing long-term treatment, effective and well-tolerated pharmacological therapy is key. Aripiprazole is a third-generation atypical antipsychotic indicated for SZ that is one of the most well tolerated and with the least discontinuation in patients [20,21]. It acts as a partial agonist of the D2 dopaminergic receptor and a full antagonist of the 5-HT_2A_ serotoninergic receptor [22]. Interestingly, the effects of aripiprazole on the aforementioned EEG features in SZ patients or animal models remain largely ill-defined. Yet, characterizing its effects on these translational parameters as well as on behavioral features would provide useful insights to better decipher its mechanism of action, which would offer valuable translational knowledge to better shape future therapeutic strategies. 

Hence, the goal of this study was to evaluate the effects of aripiprazole on the gamma-band ASSR, spontaneous gamma oscillations and behavioral impairments in an acute SZ rat model induced through pharmacological blockade of NMDAr to match the NMDAr hypofunction hypothesis of SZ. Indeed, following an acute administration of NMDAr antagonists such as phencyclidine (PCP), MK-801 (dizocilpine) or ketamine, rats display psychotic-like behavior such as increased locomotor activity, decreased social behavior, sustained cognitive impairment and a cortical excitation/inhibition imbalance [8,14,23,24]. Regarding behavioral manifestations specifically, NMDAr hypofunction is thought to produce further dysfunctions within mesolimbic and mesocortical dopaminergic pathways, contributing to positive symptoms (e.g., psychosis and agitation), cognitive impairment and negative symptoms, such as social withdrawal, in SZ patients and animal models [25,26]. Moreover, NMDAr hypofunction could directly contribute to social symptoms by provoking an imbalance between excitatory and inhibitory inputs, resulting in glutamate spillover from synapses and dysconnectivity within and between cortical and subcortical regions regulating social behavior [19]. In animal models based on pharmacological antagonism of NMDAr, the symptoms can be studied using various readouts such as social interaction deficits or hyperlocomotion, the latter being commonly used as a translational index of psychotic-like behavior. However, the validity of these behavioral tests has been challenged [27]. Considering the previous statements, blocking NMDAr in rats with NMDAr antagonists appears relevant to model SZ and offers an opportunity to test the atypical antipsychotic aripiprazole. 

## 2. Results

### 2.1. PCP, MK-801 and Ketamine Reduce the Phase Synchrony of the ASSR

To delineate the effects of commercially available antipsychotics, such as aripiprazole, and investigational drugs on SZ features, it is important to use robust in vivo animal models and translational readouts. To achieve our goal, we first evaluated the relevance of using a model of SZ based on acute NMDAr antagonism in order to reproduce in rats abnormalities in evoked gamma-band oscillations as seen in patients. In the first set of experiments, we acutely and subcutaneously injected three different NMDAr antagonists, namely PCP (5 mg/kg), MK-801 (0.2 mg/kg) and ketamine (20 mg/kg), into chronically EEG-implanted freely moving rats before recording the gamma-band ASSR by telemetry. We chose to test these three different NMDAr antagonists in order to compare their respective efficacy in reproducing ASSR impairments. The ASSR is usually assessed at frequencies around 40 Hz in SZ patients. In this study, it was elicited using 50-Hz auditory click trains as previous internal data showed a better ASSR at this frequency (data not shown) and previous published work demonstrated that 50-Hz ASSRs were more sensible to NMDAr hypofunction than 40-Hz ASSRs in rodents [28]. The vehicle-treated animals overall elicited a homogeneous and robust synchrony of ASSRs recorded in both the global (an area covering frontal cortical regions, such as the cingulate cortex and motor cortex) and auditory cortex (Figure 1A). In sharp contrast, the three groups of rats treated with NMDAr antagonists showed a significant reduction in the ASSR PLF (Figure 1A), representing 38% of the vehicle-treated group for the most affected group (44.4 ± 8.1 (*p =* 0.0034), 52.8 ± 8.5 (*p =* 0.0135) and 38.6 ± 5.6% of the vehicle-treated group (*p =* 0.0013) for PCP, MK-801 and ketamine in the global cortex, respectively and 40.3 ± 8.5 (*p <* 0.0001), 38.7 ± 3.2 (*p <* 0.0001) and 44.2 ± 5.9% of the vehicle-treated group (*p =* 0.0001) for PCP, MK-801 and ketamine for the auditory cortex, respectively (Figure 1B,C)). This effect appeared to be more significant in the auditory cortex. These first results indicate that acute NMDAr antagonism in rats is able to reproduce gamma-band ASSR abnormalities seen in SZ patients.

### 2.2. PCP, MK-801 and Ketamine Increase Locomotor Activity and Decrease Social Interaction

The effects of the three NMDAr antagonists were then evaluated and compared in terms of behavioral features relevant to SZ to further validate our approach.

First, an important increase in spontaneous locomotor activity was observed in all rats treated with NMDAr antagonists compared to the vehicle-treated animals (Figure 2A). Interestingly, this effect was very strong up to 90 min (i.e., until the end of the test) for PCP and MK-801 but not for ketamine, whose effect decreased strongly after 30 min but remained significant compared to the vehicle-treated animals. Overall, the hyperlocomotion over 90 min was similar for the PCP-treated and MK-801-treated rats and less intense for the ketamine-treated animals (3574 ± 216.3 (*p <* 0.0001), 3334 ± 231.5 (*p <* 0.0001) and 1612 ± 123.8 total activity counts (*p <* 0.0001) in PCP, MK-801 and ketamine, respectively, versus 286.8 ± 17.8 total activity counts for the vehicle-treated group (Figure 2B). 

Then, we evaluated if the acute blockade of NMDAr could also reproduce social impairments. To do so, pairs of rats were administered with the same compounds and put in an open arena for 10 min. As expected, the vehicle-treated rats displayed a significant amount of non-aggressive social behavior, representing around 10% of the overall period of scoring. In sharp contrast, the rats treated with PCP, MK-801 or ketamine showed a strong diminution of about ~80% of social interactions compared to the vehicle-treated animals (4.8 ± 0.7 (*p <* 0.0001), 13.9 ± 1.6 (*p <* 0.0001) and 10.4 ± 1.0 s of social interaction (*p <* 0.0001) for PCP, MK-801 and ketamine, respectively, compared to 57.0 ± 3.5 s of social interaction for the vehicle-treated group (Figure 2C)). 

Of note, the pre-treatment times used for the social interaction experiments (i.e., 45 min before the test for PCP and ketamine and 240 min before the test for MK-801) were chosen based on the available literature [29,30]. Unfortunately, the PCP-treated animals displayed strong motor impairments and stereotypies during the social interaction test (seen during the test and reflected in Figure 2A), which could potentially have introduced an experimental bias into the analysis of social interaction (Figure 2C). Hence, we then wanted to verify that the decrease in social behavior in the NMDAr-antagonist-treated animals was not influenced by motor defects caused by an NMDAr blockade. We investigated this aspect by focusing on MK-801, as we considered this molecule to perform experiments that evaluated aripiprazole’s effects in the model (see next sections). To do so, we measured spontaneous locomotor activity in rats following MK-801 administration up to 240 min post-administration, which corresponds to the beginning of the social interaction task for this compound. As seen in Figure 2D, the MK-801-treated animals were no more hyperactive compared to the controls from ~220 min after MK-801 administration and until the end of the recording (i.e., until 240 min). Moreover, these rats did not display any more stereotypies during the social interaction test, which confirms that the social interaction deficit seen 240 min after MK-801 administration is not influenced by any motor impairment. 

Overall, PCP (5 mg/kg), MK-801 (0.2 mg/kg) and ketamine (20 mg/kg) were able to reproduce key features of SZ, scrambling evoked gamma-band oscillations, eliciting increased agitation/psychotic-like behavior while reducing social interaction. 

Following these results, we decided to select one NMDAr antagonist for the next set of experiments aiming to characterize the effects of aripiprazole on the electrophysiological and behavioral features of the model. The choice of continuing with only one NMDAr antagonist was first motivated by ethical reasons, as evaluating aripiprazole using a single NMDAr antagonist allowed us to reduce the number of animals used. However, investigating the effects of aripiprazole on all the NMDAr antagonists remains of interest, as this could lead to valuable scientific knowledge. Thus, our choice was to continue with MK-801 due to its better potency towards NMDAr and wider commercial availability compared to the two other drugs. Moreover, the effect of MK-801 on locomotor activity has been shown in our study to not overlap with its effects on social interaction, which has not been investigated for PCP and ketamine. 

### 2.3. Aripiprazole Has No Effect on Changes Provoked by MK-801 on Evoked Gamma Oscillations

In order to know if the atypical antipsychotic aripiprazole can reverse the ASSR deficit described previously, we treated the same telemetered rats with both MK-801 and aripiprazole before initiating the ASSR paradigm. We administered aripiprazole intraperitoneally and 45 min before initiating the ASSR at a commonly used dose of 3 mg/kg because these settings were previously shown to induce behavioral and electrophysiological effects in similar rodent models of SZ [31,32,33]. Of note, this dose of 3 mg/kg in rats corresponds to ~30 mg in humans, which is within the therapeutic dose range used in clinical studies [34,35,36,37]. As shown in Figure 3, an acute administration of MK-801 was again able to elicit a strong deficit of ~50% in the PLF of the ASSR on both the global cortex and auditory cortex channel (48.2 ± 6.8 (*p =* 0.0048) and 51.2 ± 6.8% of the vehicle-treated group (*p =* 0.0239) for global and auditory cortex, respectively (Figure 3B,C)). However, no effect of aripiprazole (3 mg/kg) could be seen on this deficit (Figure 3). 

### 2.4. Aripiprazole Does Not Normalize the Increase in Spontaneous Gamma Oscillations Provoked by MK-801

Another very common electrophysiological feature following NMDAr blockade and present in SZ [6,9], but that was not characterized in our model in the first set of experiments, is the increased spontaneous gamma power in the EEG. We took advantage of these new experiments to evaluate if the MK-801-treated rats also displayed increased spontaneous gamma power and if aripiprazole was able to normalize this parameter. To achieve this goal, rat EEG was recorded under normal conditions (i.e., without any auditory stimulation) for 1 h following the ASSR paradigm. As shown in Figure 4, MK-801 significantly increased the spontaneous gamma power compared to the control group (406.2 ± 76.7 (*p =* 0.0058) and 255.3 ± 32.18% of the vehicle-treated group (*p =* 0.0143) in global and auditory cortex, respectively). Here again, we observed no effect of aripiprazole on the increased gamma-band activity induced by MK-801, either in the global cortex or auditory cortex channel (Figure 4).

### 2.5. Aripiprazole Alleviates MK-801-Induced Hyperactivity but Not Social Deficit

After completing the electrophysiological studies, we finally tested the effect of aripiprazole on MK-801-induced hyperactivity, as it is known in the literature to reduce this behavioral parameter in rats, which reflects its efficacy against positive symptoms [38]. Doses of 1 to 30 mg/kg of aripiprazole were chosen based on previous studies reporting behavioral effects mediated by this drug at comparable doses [32,37,39]. As shown in Figure 5A, an acute intraperitoneal injection of aripiprazole indeed significantly reduced MK-801-induced hyperactivity. This effect was moderate for the two highest doses of aripiprazole (10 and 30 mg/kg) throughout the first part of the recording (i.e., from 0 to 80 min of recording) and was robust and significant from 80 to 150 min following MK-801 administration, where we could demonstrate that doses of 3, 10 and 30 mg/kg induced a dose-dependent reduction in MK-801-induced hyperactivity (1162 ± 159.8 (*p =* 0.0675), 954.7 ± 147.1 (*p =* 0.0048), 799.4 ± 151.3 (*p =* 0.0004) and 811.8 ± 106.6 activity counts (*p =* 0.0003) from 80 to 150 min in the MK-801-treated animals also administered with aripiprazole at 1, 3, 10 and 30 mg/kg, respectively, compared to 1629 ± 170.0 activity counts for the MK-801-treated group (Figure 5B)). We also acutely treated a group of animals with clozapine, another antipsychotic drug with known effects on positive-like symptoms [40], which was used here as a reference item. As expected, a strong reduction in MK-801-induced hyperlocomotion was observed with clozapine (10 mg/kg) in the first two hours following MK-801 administration (1456 ± 301.8 activity counts (*p =* 0.0011) in the MK-801/clozapine-treated animals compared to 3081 ± 409.1 activity counts for the MK-801/vehicle-treated rats over the 0–120 min period (Figure 5A,C)).

Lastly, we wanted to assess aripiprazole’s effect on MK-801-induced social deficit in the rats. While MK-801 was shown to significantly impair social interaction compared to the vehicle-treated animals (24.8 ± 4.0 s of social interaction (*p <* 0.0001) for the MK-801-treated group compared to 62.7 ± 7.2 s of social interaction for the vehicle-treated group), an acute and intraperitoneal injection of aripiprazole at the three doses tested (1, 3 and 10 mg/kg) had no effect on the deficit (Figure 6). 

## 3. Discussion

In this paper, we provide a broad characterization of the effects of aripiprazole on SZ-like features, such as abnormal evoked and spontaneous gamma oscillations, hyperlocomotion and social withdrawal, in an acute SZ rat model induced by NMDAr antagonism. 

First, we characterized the model in which aripiprazole was further tested. This model displays several relevant features and was induced using three different NMDAr antagonists (PCP, MK-801 and ketamine). We first highlighted in telemetered rats that the NMDAr antagonists reduced the phase synchrony of evoked gamma oscillations by using the gamma-band ASSR, which is reflective of an abnormal cortical excitation/inhibition balance, an important translational neurophysiological feature of SZ that is associated with all symptoms observed in patients [15,16,17]. Among those symptoms, we also assessed in rats positive-like symptoms through hyperlocomotion evaluation and negative-like symptoms through the characterization of a deficit in the social interaction test. These first results perfectly align with previous experiments showing that an acute administration of NMDAr antagonists can reproduce the gamma-band ASSR [14] and social interaction abnormalities [23], as well as a sustained hyperlocomotion [24]. Although the reductions in the ASSR PLF and social behavior were similar for the three NMDAr antagonists in our study and these molecules have a lot of characteristics in common, it is still worth noting that they also display slight differences. For instance, ketamine is less potent towards NMDAr compared to PCP and MK-801, and these three NMDAr antagonists also display different affinities towards each subunit of the NMDAr [41,42]. These aspects might explain why previous studies found that different neurochemical and behavioral profiles could be induced by these three compounds [32,41,43] and why ketamine produced a weaker hyperlocomotion and was used at a higher dose compared to the two other compounds in our study. Overall, our results also resonate with the clinical situation, making the model useful, translational and robust to evaluate commercially available antipsychotics and new drug candidates indicated for SZ.

In pursuing our goal of characterizing the effect of aripiprazole on the electrophysiological and behavioral features of the model, we next found that a common dose of 3 mg/kg of this atypical antipsychotic had no effect on the abnormalities seen in the ASSR after acute NMDAr antagonism. Since only a single dose of aripiprazole was tested, it cannot be ruled out that aripiprazole could still rescue the ASSR at higher or lower acute doses in our model, or even following chronic administrations, which deserves to be further tested. To the best of our knowledge, this study is the first to report aripiprazole’s absence of effect on the ASSR in a rat model of SZ. It is also worth noting that our work aligns with two clinical studies that also found no difference in the gamma-band ASSR between atypical-antipsychotic-treated and untreated SZ patients [44,45]. However, two other studies found that atypical antipsychotics could alleviate the deficit in evoked gamma oscillations that SZ patients exhibit [46,47]. Unfortunately, these contradictory results cannot be easily explained, as the latter studies cited either did not mention the names of the drugs that were used or included various atypical antipsychotic treatments in their treatment arm. As the mechanisms of action are not the same among all atypical antipsychotics, these inconsistent literature reports clearly highlight the need to further investigate the effects of each antipsychotic independently on evoked gamma oscillations in order to fully understand their individual effects on this parameter. 

After investigating the effect of aripiprazole on evoked gamma-band oscillations using the ASSR paradigm, we then looked at its effects on spontaneous gamma oscillations, which is also a relevant neurophysiological feature of SZ and associated with the NMDAr-hypofunction-mediated excitation/inhibition imbalance [6]. While we showed that MK-801 was able to increase spontaneous gamma power in rats, reproducing a similar phenomenon occurring in SZ patients and animal models [6], we have again demonstrated for the first time that there is no clear effect of aripiprazole at a dose of 3 mg/kg on this parameter. Interestingly, a few studies have investigated the effects of other atypical antipsychotics on this parameter. Aligning with our results, two clinical studies concluded that typical- and atypical-antipsychotic treatments had no effect on spontaneous gamma oscillations in SZ patients [48,49]. Likewise, two animal studies demonstrated that the atypical antipsychotic clozapine had no effect on the increase in spontaneous gamma oscillations caused by a pharmacological blockade of NMDAr in rats [50,51]. Although our results tend to align with previous work carried out in patients and animal models, it is worth noting that different outcomes have been found in vivo for clozapine and olanzapine [52]. Similarly, ex vivo studies using hippocampal and prefrontal cortex slices subjected to MK-801 and treated with cariprazine and clozapine showed no improvement in spontaneous gamma oscillations [53,54]. Again, these contradictory results clearly highlight the need to better characterize the individual effects of each atypical antipsychotic on gamma oscillations in SZ patients and relevant models of the disease because of their different mechanisms of action. As suggested previously, understanding the individual effects of available antipsychotics in SZ patients or animal models and on relevant translational EEG markers such as evoked and spontaneous gamma oscillations may help to understand which effect is to be expected for a new drug to demonstrate efficacy against these aberrant electrophysiological features and their associated symptoms. 

Furthermore, the use of behavioral tests coupled with electrophysiological assessments appear relevant to better understand the mechanisms of action of antipsychotics and investigational drugs. In this respect, our results highlight that aripiprazole had no effect on the social interaction deficit induced by MK-801 while it was still able to reduce MK-801-induced hyperactivity. Consistently, it has been shown that aripiprazole and other atypical antipsychotics have no clinically meaningful effects on negative symptoms, including social withdrawal, while they can still effectively alleviate positive symptoms of SZ [1,2,3,55,56,57]. In fact, although atypical antipsychotics like aripiprazole may still induce statistically significant improvements in indices of social behavior in patients, which is not the case for first-generation antipsychotics, these are described as too small to produce a real improvement in patients’ quality of life. Furthermore, these moderate effects might result from a reduction in positive symptoms, which in turn would improve patients’ overall well-being and indirectly increase social functioning [2,55,57]. An influential clinical meta-analysis even demonstrated that aripiprazole had no superiority compared to first-generation antipsychotics in the treatment of social dysfunction and other negative symptoms [58]. While aripiprazole and other atypical antipsychotics might thus not directly target the root cause of social dysfunction, it has even been suggested that their adverse effects, such as sedation, may interfere with the clinical outcome [2,55]. From this perspective, it is important to note that in our study, some animals treated with aripiprazole at all doses (1 to 10 mg/kg) occasionally displayed some sedation signs during the social interaction test, which potentially might have prevented us from seeing a potential improvement in this test. This aspect may also explain why the literature about the effects of aripiprazole and other antipsychotics on social behavior in SZ animal models based on NMDAr hypofunction is inconclusive. One rat study, for instance, found that aripiprazole administered at lower doses than those used in our study could reverse the social interaction deficit induced by an acute administration of MK-801 [59]. On the other hand, other studies found that aripiprazole used at comparable doses as well as other atypical antipsychotics such as olanzapine, risperidone and clozapine had no effect on the social deficit provoked by MK-801 or PCP [32,60,61,62], which perfectly aligns with our results. Hence, it is possible that the lack of effect of aripiprazole was either because the dose tested was too strong or because it was not able to produce an effect on the root cause of social dysfunction. In fact, aripiprazole was also unable to recover the abnormalities found in evoked and spontaneous gamma oscillations in the model, two EEG parameters measuring aspects of the neuronal excitation/inhibition imbalance that underlies social dysfunction in SZ [8,9,14,18,19]. Thus, this suggests that aripiprazole’s inefficacy towards social impairment in our study and its very limited effects against social deficits in SZ patients might be associated with (and potentially a consequence of) its inefficacy towards the aforementioned electrophysiological impairments that contribute to these symptoms. 

Considering the above, our study clearly underlines the importance of (i) targeting the key neural mechanisms of SZ pathophysiology, such as NMDAr hypofunction, to reduce symptoms that depend on these key features and (ii) combining translational electrophysiological readouts, such as EEG biomarkers, with behavioral measures to obtain a broader picture of the pharmacological activity of an investigational drug indicated for SZ. This strategy may also be relevant for other symptoms of SZ such as cognitive impairment, which is known to be associated with gamma-band abnormalities too [6,63]. Compounds for which drug developers have followed this strategy have yielded promising results on electrophysiological measures as well as indices of social and cognitive functioning in patients and animal models. For instance, drugs increasing the synaptic concentration of the NMDAr co-agonists glycine and D-serine or acting on other aspects of the glutamatergic synapse are showing promising results on EEG and behavioral measures in preclinical and clinical trials for social and cognitive symptoms of SZ [64,65,66,67,68]. 

## 4. Materials and Methods

### 4.1. Animals

All experiments were performed on male Sprague Dawley rats (Janvier Labs, C.S. 4105, Le Genest-Saint-Isle, Saint Berthevin, France), weighing 175–200 g at reception. All rats were housed under standard and controlled conditions: room temperature (22 ± 2 °C), hygrometry (55 ± 10%), light/dark cycle (12 h/12 h, light on from 8:00 a.m. to 8:00 p.m.), air replacement (15–20 volumes/hour), water and food (SAFE, ref. A04) ad libitum. All rats were allowed to habituate to environmental conditions for at least 5 days prior to each experiment. Three batches of animals were used in this study: Batch A: animals used for both EEG experiments (n = 12 implanted rats weighing between 207 and 293 g on the day of surgery).Batch B: animals used for the behavioral experiments described in Section 2.2 (n = 40 rats weighing between 226 and 298 g on the first day of the locomotor activity evaluation and between 240 and 314 g on the day of the social interaction evaluation) (Figure 2).Batch C: animals used for the behavioral experiments described in Section 2.5 (n = 70 rats weighing between 291 and 444 g on the first day of the locomotor activity evaluation and between 404 and 508 g on the day of the social interaction evaluation) (Figure 5 and Figure 6).

#### 4.1.1. Batch A: Animals Used for EEG Experiments

Prior to the implantation surgery, the rats were housed in groups of 2–4 in polysulfone cages (floor area = 1500 cm^2^). Chronically instrumented rats were then individually housed in polysulfone cages (floor area = 1500 cm^2^) after surgery until the end of the study with enrichment and were identified by an ear tattoo. 

#### 4.1.2. Batches B and C: Animals Used for Behavioral Experiments

The animals were housed individually throughout the experimental phase in polysulfone cages (floor area = 1500 cm^2^) to promote social behavior during the social interaction test. The rats were numbered in each cage by marking their tail with indelible markers. Animals were randomized to each experimental group prior to locomotor activity evaluation (actimetry). Then, after a sufficient wash-out period of at least one week, the same rats were randomized again and allocated to different treatment groups for the social interaction test. After a sufficient wash-out period of at least one week, some rats from batch B were re-randomized and re-used a third time to complete the spontaneous locomotor activity evaluation performed over 240 min that is described in Section 2.2 and Figure 2D. All animals were identified by marking their tail with indelible markers. 

### 4.2. Drugs and Dosing Schemes for Each Test

#### 4.2.1. Drugs

All drugs were freshly prepared prior to the administration. PCP (Sigma, P3029), MK-801 (Carbosynth, BM162595) and ketamine (Imalgene 1000, Centravet, IMA004) were prepared in NaCl 0.9% (Osalia) under magnetic stirring until a solution was obtained. Aripiprazole (MedChemExpress, HY-14546) was prepared in Tween^®^ 80 10% (Sigma, P8074) and NaCl 0.9% under magnetic stirring until a solution was obtained. Clozapine (Carbosynth, FC20526) was prepared in HCl 0.1 M (Sigma, 2104-50ML) 5% and NaCl 0.9% under magnetic stirring until a solution was obtained.

#### 4.2.2. Dosing Schemes for Each Test

Due to the specificity of each test, the compounds were administered at different doses, routes and timings of administration, which are summarized in Table 1 for the EEG experiments, Table 2 for the locomotor activity experiments and Table 3 for the social interaction tests. Compounds were given through subcutaneous (sc) or intraperitoneal (ip) route.

### 4.3. Surgical Implantation of the Telemetry Transmitter

Surgery was performed under isoflurane anesthesia (5% isoflurane/air for anesthesia induction and 1.5–3% isoflurane/air for anesthesia maintenance) and proper anti-inflammatory and analgesic treatment (meloxicam 1–2 mg/kg, sc and lidocaine 2.1 mg/mL, 100–200 µL/ operated site, sc). Then, the animal was placed in a stereotaxic framework (David Kopf Instrument, Tujunga, CA, USA). The four EEG leads (made of a nickel/cobalt-based alloy) of the telemetry implant (HD-S02 implant, Data Sciences International, St Paul, MN, USA) were fixed on the skull with screws and dental acrylic at the following stereotaxic coordinates [69]: -Channel #1, global cortex: AP: +2.0 mm and ML: −1.5 mm from bregma for the active electrode, which is therefore located at the level of the frontal cortex, with a reference on the right cerebellum, i.e., AP: −11.0 mm and ML: −3.0 mm from bregma.-Channel #2, auditory cortex: AP: −4.8 mm and ML: +7 mm from bregma for the active electrode, which is at the level of the auditory cortex, with a reference on the left cerebellum, i.e., AP: −11.0 mm and ML: +3.0 mm from bregma.

The telemetry transmitter itself was placed subcutaneously along the animal’s flank and attached with non-absorbable sutures. The skin incisions were then closed with re-absorbable sutures.

Post-surgical analgesia was ensured by subcutaneous administrations of buprenorphine (10–50 µg/kg, sc) twice a day for 2 days (including surgery day) and meloxicam (1–2 mg/kg, sc) once a day for 3 days (including surgery day). At least 7 days were allowed to ensure the complete recovery from the surgery before the first EEG experiment. During this recovery period, the animals were weighed and observed daily, and the sutures were examined and disinfected if needed with an antiseptic solution (povidone-iodine).

The quality control of electrophysiological signals (visual inspection of the signal and spectral analysis of the EEG on a short recording) was performed a few days prior to the first dosing/recording to select animals to be included in the study.

### 4.4. EEG Recording 

All electrophysiological recordings took place between 9:00 a.m. and 1:30 p.m. (during the light phase). 

The telemetry data was collected using an acquisition system from Data Sciences International (St Paul, MN, USA). During the experimental session, the animals were individually housed in cages placed on a telemetry receiver panel. EEG signals were monitored and stored on the hard drive of a computer using the HEM software (version 4.4, Notocord, France).

All results were obtained from electrophysiological recordings performed on the same instrumented animals by using Latin square designs. A first set of 4 recordings was performed through a first Latin square design aiming to characterize the effects of the NMDAr antagonists PCP, MK-801 and ketamine on the ASSR. Then, a second set of 3 recording sessions took place to assess the effects of aripiprazole on MK-801-induced changes in the ASSR and spontaneous gamma oscillations. A recording session consisted of a first 30-min ASSR paradigm followed by a 1-h recording of spontaneous cortical gamma oscillations, which are further described in Section 4.4.1 and Section 4.4.2. A wash-out period of at least 2 days between two recording/dosing sessions was applied.

#### 4.4.1. ASSR Recording

First, each recording session started by recording EEG activity in the gamma range in response to auditory stimuli. The EEG was continuously recorded during a 30-min session with acoustic stimulations, which consisted of a set of 1500 stimuli. A stimulus set was made of a 500-ms click train followed by a 700-ms silence, finally corresponding to 50-Hz stimuli. 50-Hz stimuli were chosen instead of 40-Hz stimuli (which are classically used in human ASSR studies) based on previous internal results showing a better ASSR at this frequency (data not shown) and previous work suggesting that ASSR PLF deficits induced by NMDAr antagonists can be better captured at gamma frequencies around 50 Hz [28]. Stimuli were calibrated using a sound calibrator (Laserliner SoundTest Master, Germany) before each acquisition to adjust the sound level to 79 dB.

#### 4.4.2. Recording of Spontaneous EEG for qEEG Analysis

Spontaneous EEG (without any stimulation) was continuously recorded for 1 h immediately after the end of the ASSR stimulation protocol for qEEG analyses.

### 4.5. EEG Signal Processing

#### 4.5.1. ASSR Analysis

EEG signal processing was performed using Matlab (Mathworks, Inc., Natick, MA, USA) scripts. The continuous EEG was split into 1.5-s signal segments (i.e., epochs) starting 500 ms prior to the stimulus onset ([−500, +1000] ms around the trigger). For each epoch, the median on the whole epoch was removed from whole epoch values (offset correction). Automatic epoch rejection relies on the threshold method: if the amplitude of any sample of an epoch exceeds +/−300 μV, this epoch is rejected from analysis.

To determine the PLF, time–frequency representations (TFRs) were calculated for each epoch on [−500, +1000] ms × [48; 52] Hz areas for the 50-Hz stimulation. TFRs were computed using 9-cycle Morlet wavelets. Normalized amplitude TFRs were averaged to estimate PLF TFRs. On those TFRs, the baseline (BL) was defined as the area [−500, 0] ms × [48; 52] Hz, and the post-stimulus region of interest (ROI) is the area [0, 500] ms × [48; 52] Hz. The selected time intervals were shortened by the half length of the wavelet (94 ms for 9 cycles at 48 Hz; rounded to 100 ms). The PLF parameters exported for analysis are the value on the ROI minus the value on the BL.

#### 4.5.2. qEEG Analysis

The qEEG analysis was performed with Matlab (Mathworks, Inc., Natick, MA, USA). The spectral analysis of the EEG was performed on non-overlapping 4-s epochs using a fast Fourier transform (FFT) and Hanning window. Epochs with a peak-to-peak amplitude higher than 400 µV were excluded from the next steps of the analysis.

Each spectrum has the same resolution (0.25 Hz) and the same frequency range (0–500 Hz). A FFT provides complex spectra. Power spectra were obtained by using the product of the complex spectra by the complex conjugate spectra. From each spectrum, the power in the gamma band was extracted (sum of spectrum values within the band 32–80 Hz).

The effect of the different molecules on the gamma band of the EEG signal was evaluated during the 1-h period of recording where no acoustic stimulation was performed. 

### 4.6. Behavioral Evaluation

#### 4.6.1. Locomotor Activity Evaluation (Actimetry)

Animals were placed individually into cages positioned in an activity meter (Imetronic system) for a period of up to 240 min, starting immediately after administration and between 2:00 and 6:00 p.m. This apparatus uses infrared beams to record horizontal animal movements. The animals had no access to food or water during the recording period.

#### 4.6.2. Social Interaction Evaluation

To evaluate the level of social interaction following dosing, rats were tested in the social interaction test. Couples for the social interaction test were matched by treatment groups. After each member of a couple received the treatment, the couples were put for 10 min in an open field (57 cm × 67 cm × 30 cm) placed under dim light. The time spent in active, non-aggressive social behavior by a couple of rats was then scored for both rats. Active social behavior was defined as sniffing, following, grooming, kicking, mounting, jumping on, wrestling, boxing and crawling under and over the partner.

After each session, the rats were returned to their home cage, the fecal boli were removed, and the arena’s walls were cleaned with a 70% ethanol solution.

### 4.7. Statistical Analyses

All data sets were cleared from individual outliers displaying aberrant electrophysiological or behavioral responses compared to the other animals in their group by removing the data being beyond two standard deviations.

For the first set of results characterizing the effects of NMDAr antagonists on the ASSR, locomotor activity and social interaction (Section 2.1 and Section 2.2; Figure 1 and Figure 2), a one-way ANOVA was used to analyze the data. Dunnett’s post hoc analysis was then performed to assess differences against the vehicle-treated group. 

For the second set of results aiming at characterizing the effects of aripiprazole on the changes produced by MK-801 on the ASSR, spontaneous gamma oscillations, locomotor activity and social interaction (Section 2.3, Section 2.4 and Section 2.5; Figure 3, Figure 4, Figure 5 and Figure 6), a one-way ANOVA was used to analyze the data. Dunnett’s post hoc analysis was then performed to assess differences against the MK-801-treated group. 

## 5. Conclusions

In this study, we first characterized a rat model based on NMDAr hypofunction that displays relevant behavioral and electrophysiological features, making it well-suited to test aripiprazole on measures of interest. When administered acutely, the NMDAr antagonists PCP, MK-801 and ketamine were indeed able to alter the gamma-band ASSR phase synchrony, spontaneous gamma oscillations and social interaction and produce a sustained hyperactivity, the latter being used as an index of increased psychotic-like behavior. We then showed that aripiprazole at a commonly used dose of 3 mg/kg could not reverse the abnormalities in evoked and spontaneous gamma-band oscillations in the model. While aripiprazole at doses ranging from 1 to 10 mg/kg could also not reverse the deficit in social interaction produced by MK-801, this atypical antipsychotic was still able to reduce MK-801-induced hyperactivity at doses ranging from 3 to 30 mg/kg. Aligning with the body of literature suggesting that atypical antipsychotics such as aripiprazole do not have clinically meaningful effects on negative symptoms while still being able to reduce positive symptoms, our results are the first to describe the lack of effect of this drug on evoked and spontaneous gamma oscillations in a rat model of SZ. Although a single dose of 3 mg/kg was investigated on EEG readouts in this study, the lack of effect of aripiprazole on these may suggest the inability of this drug to rescue gamma-band abnormalities and the cortical excitation/inhibition imbalance produced by NMDAr hypofunctioning in our model. Because social impairment depends strongly on gamma oscillations and NMDAr functioning, it is also possible that aripiprazole’s inefficacy against social dysfunction in our model and its limited efficacy on patients’ social withdrawal are associated with (and potentially a consequence of) the inability of this drug to normalize gamma oscillations and the neuronal excitation/inhibition imbalance that underly social dysfunction. Understanding this mechanism with a combination of translational EEG measures directly probing the key neural mechanisms of SZ and behavioral readouts may therefore help to better characterize already commercialized antipsychotics and investigational drugs for higher chances of success in SZ drug development. 

## Figures and Tables

**Figure 1 ijms-25-01035-f001:**
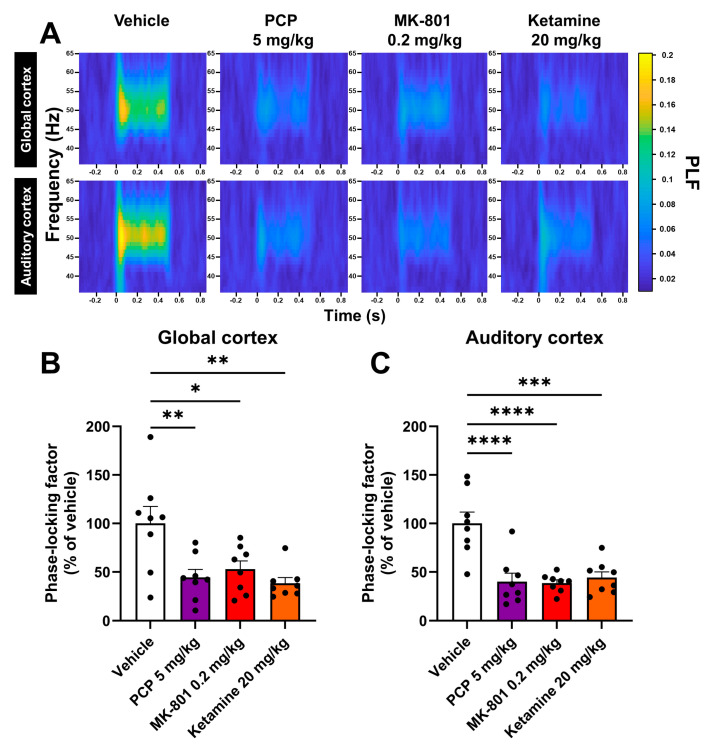
An acute and subcutaneous injection of PCP, MK-801 or ketamine reduced the phase synchrony of the ASSR in rats. (**A**) Time-frequency representations of the average PLF of the gamma-band ASSR recorded in rats’ global and auditory cortex following an acute subcutaneous administration with vehicle or the NMDAr antagonists PCP, MK-801 and ketamine. (**B**) Histogram showing the average percentage of change from the vehicle-treated group for the PLF of the gamma-band ASSR recorded in rats’ global cortex following an administration with vehicle or the NMDAr antagonists. (**C**) Histogram showing the average percentage of change from the vehicle-treated group for the PLF of the gamma-band ASSR recorded in rats’ auditory cortex following an administration with vehicle or the NMDAr antagonists. Results presented as means ± SEM with individual values (n = 8/group), * *p* ≤ 0.05, ** *p* ≤ 0.01, *** *p* ≤ 0.001, **** *p* ≤ 0.0001 vs. vehicle-treated group using a one-way ANOVA followed by Dunnett’s post hoc test.

**Figure 2 ijms-25-01035-f002:**
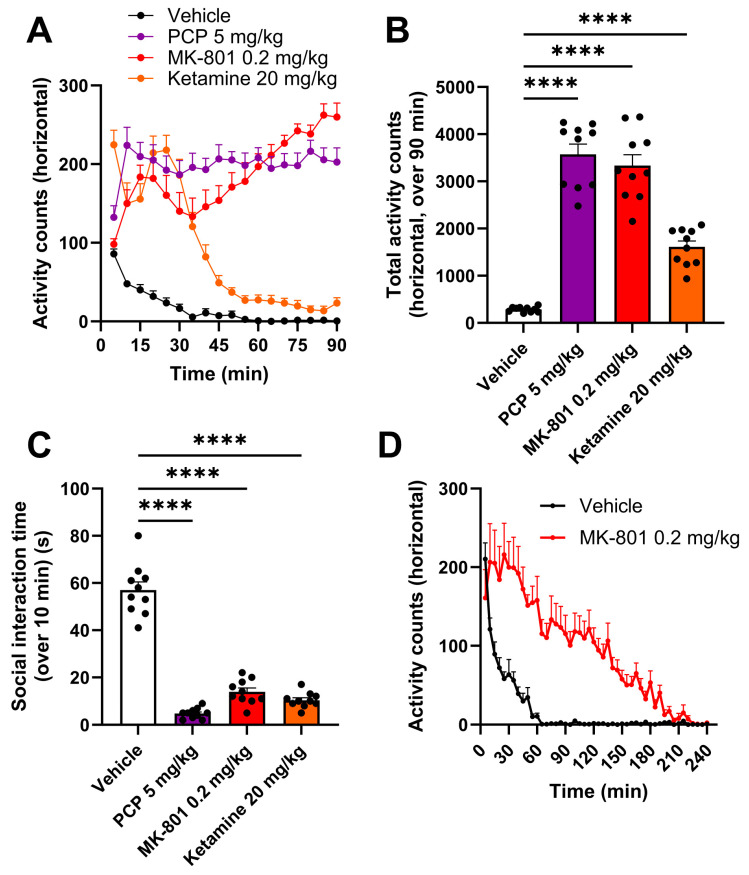
An acute and subcutaneous injection of PCP, MK-801 or ketamine increased spontaneous locomotor activity and reduced social behavior in rats. (**A**) Horizontal activity plotted over 90 min following acute subcutaneous administration in rats of PCP, MK-801 or ketamine in comparison with vehicle-treated animals. (**B**) Horizontal activity summed over the 90-min period of recording following an acute subcutaneous administration in rats with PCP, MK-801 or ketamine in comparison with vehicle-treated animals. (**C**) Social interaction time over the 10-min test in rats treated with PCP, MK-801 or ketamine in comparison with vehicle-treated animals. (**D**) Horizontal activity plotted over 240 min following an acute subcutaneous administration with MK-801 or vehicle in rats. Results presented as means ± SEM with individual values (n = 10/group), **** *p* ≤ 0.0001 vs. the vehicle-treated group using a one-way ANOVA followed by Dunnett’s post hoc test.

**Figure 3 ijms-25-01035-f003:**
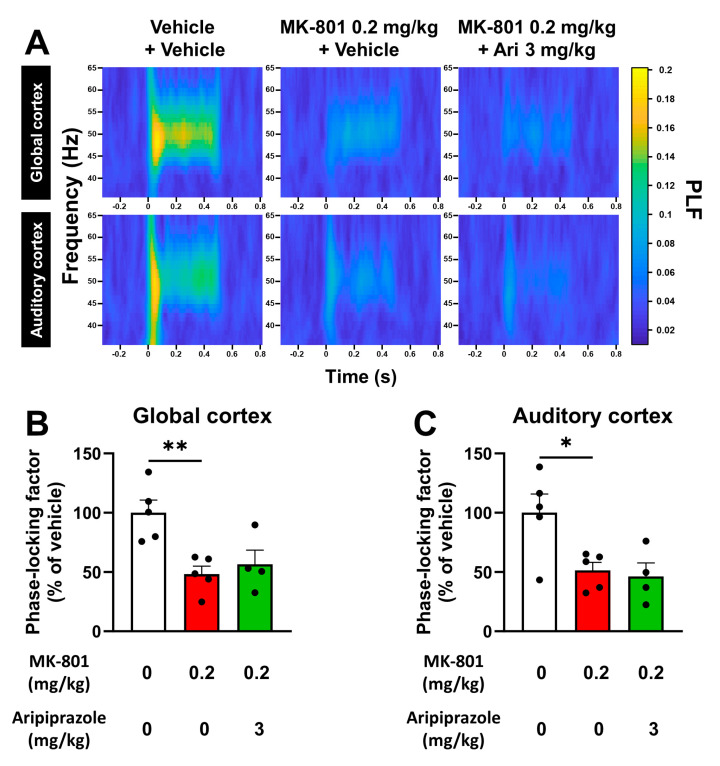
Aripiprazole administered acutely and intraperitoneally does not recover the ASSR deficit produced by MK-801 in rats. (**A**) Time-frequency representations of the average PLF of the gamma-band ASSR recorded in rats’ global and auditory cortex following an acute administration with aripiprazole (Ari) at a dose of 3 mg/kg in MK-801-treated rats in comparison with vehicle/vehicle-treated and MK-801/vehicle-treated animals. (**B**) Histogram showing the average percentage of change from the vehicle/vehicle-treated group for the PLF of the gamma-band ASSR recorded in rats’ global cortex following an acute administration with aripiprazole and MK-801 in comparison with vehicle/vehicle-treated animals and MK-801/vehicle-treated animals. (**C**) Histogram showing the average percentage of change from the vehicle/vehicle-treated group for the PLF of the gamma-band ASSR recorded in rats’ auditory cortex following an acute administration with aripiprazole and MK-801 in comparison with vehicle/vehicle-treated animals and MK-801/vehicle-treated animals. Results presented as means ± SEM with individual values (n = 4–5/group), * *p* ≤ 0.05, ** *p* ≤ 0.01 vs. the MK-801-treated group using a one-way ANOVA followed by Dunnett’s post hoc test.

**Figure 4 ijms-25-01035-f004:**
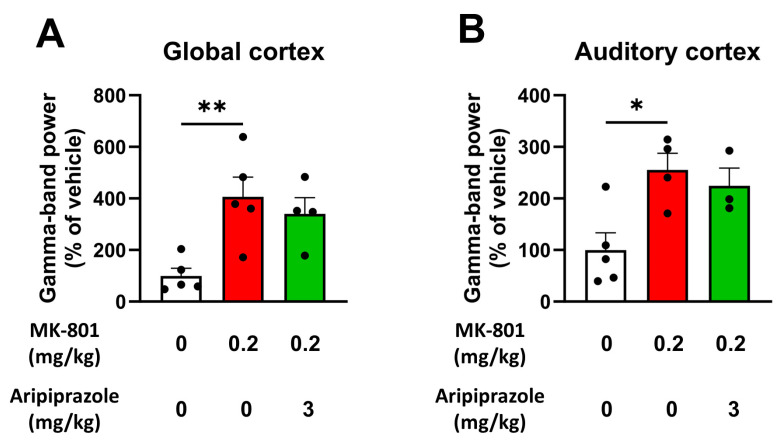
An acute and intraperitoneal administration of aripiprazole does not normalize the increase in spontaneous gamma power produced by MK-801 in rats. (**A**) Histogram showing the average percentage of change from the vehicle/vehicle-treated group for the gamma-band absolute power recorded in rats’ global cortex following an acute administration with aripiprazole and MK-801 in comparison with vehicle/vehicle-treated animals and MK-801/vehicle-treated animals. (**B**) Histogram showing the average percentage of change from the vehicle/vehicle-treated group for the gamma-band absolute power recorded in rats’ auditory cortex following an acute administration with aripiprazole and MK-801 in comparison with vehicle/vehicle-treated animals and MK-801/vehicle-treated animals. Results presented as means ± SEM with individual values (n = 3–5/group), * *p* ≤ 0.05, ** *p* ≤ 0.01, vs. the MK-801-treated group using one-way ANOVA followed by Dunnett’s post hoc test.

**Figure 5 ijms-25-01035-f005:**
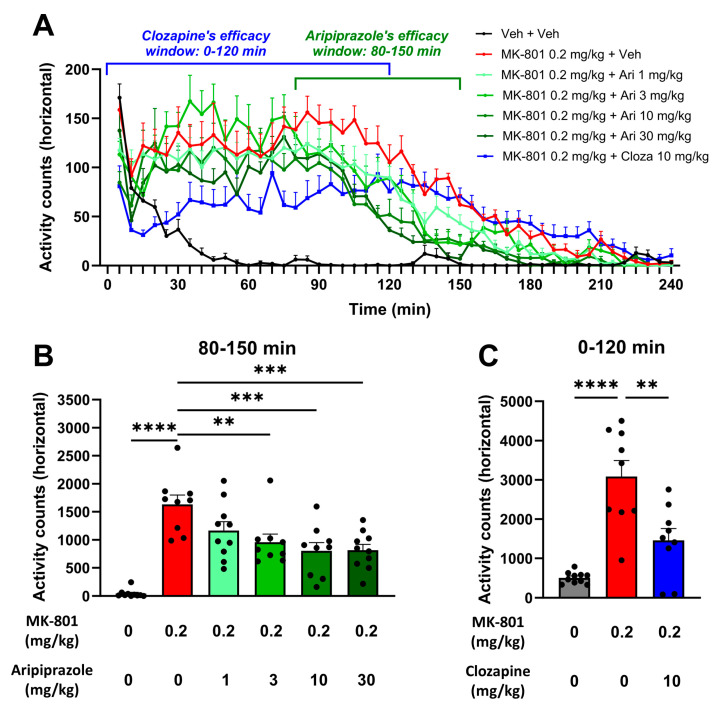
Aripiprazole administered acutely and intraperitoneally reduces MK-801-induced hyperactivity in a dose-dependent manner in rats. (**A**) Horizontal activity plotted over 240 min following an acute administration with aripiprazole (Ari) at 1, 3, 10 and 30 mg/kg in MK-801-treated rats in comparison with vehicle (Veh)/vehicle-treated, MK-801/vehicle-treated or MK-801/clozapine (Cloza)-treated rats (clozapine being used as a positive control). (**B**) Horizontal activity summed over the period between 80 min and 150 min post-recording start in MK-801-treated rats acutely injected with aripiprazole at 1, 3, 10 and 30 mg/kg in comparison with vehicle/vehicle and MK-801/vehicle-treated rats. (**C**) Horizontal activity summed over the period between 0 and 120 min post-recording start in MK-801-treated rats acutely treated with clozapine and in comparison with vehicle/vehicle-treated or MK-801/vehicle-treated rats. Results presented as means ± SEM with individual values (n = 8–10/group), ** *p* ≤ 0.01, *** *p* ≤ 0.001, **** *p* ≤ 0.0001 vs. the MK-801-treated group using a one-way ANOVA followed by Dunnett’s post hoc test.

**Figure 6 ijms-25-01035-f006:**
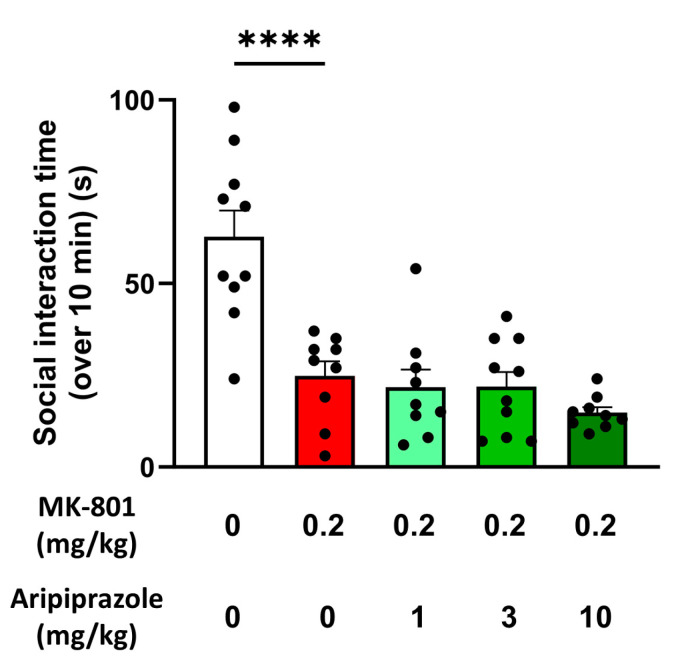
Aripiprazole does not reduce MK-801-induced social impairment in rats. Social interaction time over the 10-min test after an acute administration of aripiprazole at 1, 3 and 10 mg/kg in MK-801-treated rats in comparison with vehicle/vehicle-treated and MK-801/vehicle-treated rats. Results presented as means ± SEM with individual values (n = 9–10/group), **** *p* ≤ 0.0001 vs. the MK-801-treated group using a one-way ANOVA followed by Dunnett’s post hoc test.

**Table 1 ijms-25-01035-t001:** Compounds, doses and pre-treatment times used for EEG experiments.

Compound	Dose, Route and Volume of Administration	Time of Administration
PCP	5 mg/kg, sc (2 mL/kg)	15 min before the ASSR paradigm
MK-801	0.2 mg/kg, sc (2 mL/kg)	15 min before the ASSR paradigm
Ketamine	20 mg/kg, sc (2 mL/kg)	15 min before the ASSR paradigm
Vehicle of PCP, MK-801 and ketamine (NaCl 0.9%)	sc (2 mL/kg)	15 min before the ASSR paradigm
Aripiprazole	3 mg/kg, ip (2 mL/kg)	45 min before the ASSR paradigm
Aripiprazole’s vehicle	ip (2 mL/kg)	45 min before the ASSR paradigm

**Table 2 ijms-25-01035-t002:** Compounds, doses and pre-treatment times used for actimetry experiments.

Compound	Dose, Route and Volume of Administration	Time of Administration
PCP	5 mg/kg, sc (2 mL/kg)	Just before actimetry
MK-801	0.2 mg/kg, sc (2 mL/kg)	Just before actimetry
Ketamine	20 mg/kg, sc (2 mL/kg)	Just before actimetry
Vehicle of PCP, MK-801 and ketamine (NaCl 0.9%)	sc (2 mL/kg)	Just before actimetry
Aripiprazole	1, 3, 10, 30 mg/kg, ip (2 mL/kg)	30 min before actimetry
Aripiprazole’s vehicle	ip (2 mL/kg)	30 min before actimetry

**Table 3 ijms-25-01035-t003:** Compounds, doses and pre-treatment times used for social interaction experiments.

Compound	Dose, Route and Volume of Administration	Time of Administration
PCP	5 mg/kg, sc (2 mL/kg)	45 min before the social interaction test
MK-801	0.2 mg/kg, sc (2 mL/kg)	240 min before the social interaction test
Ketamine	20 mg/kg, sc (2 mL/kg)	30 min before the social interaction test
Vehicle of PCP, MK-801 and ketamine (NaCl 0.9%, used in Section 2.2)	sc (2 mL/kg)	30 min before the social interaction test
MK-801’s vehicle (used in Section 2.5)	sc (2 mL/kg)	240 min before the social interaction test
Aripiprazole	1, 3, 10 mg/kg, ip (2 mL/kg)	30 min before the social interaction test
Aripiprazole’s vehicle	ip (2 mL/kg)	30 min before the social interaction test

## Data Availability

Raw data can be requested with sufficient reasons.

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
