# Peer review of "Differential Effects of Aripiprazole on Electroencephalography-Recorded Gamma-Band Auditory Steady-State Response, Spontaneous Gamma Oscillations and Behavior in a Schizophrenia Rat Model"

_ijms, 2024, doi:10.3390/ijms25021035_

Round 1

Reviewer 1 Report

Comments and Suggestions for Authors

Dear Authors,

The article Differential Effects of Aripiprazole on EEG-recorded Gamma-2 Band Auditory Steady-State Response, Spontaneous Gamma Oscillations and Behavior in a Schizophrenia Rat Model” introduces in vivo study of the antipsychotic drug Aripiprazole and its effects compared with the NMDA receptor blockers.

   The article is well-written; all the details of the study are carefully illustrated.

   My proposal, as the NMDA receptor antagonists have been used in the study, it is desirable to highlite their influence on Aripiprazole`s  pharmacological action and to discuss about NMDA receptor role in Aripiprazole` mechanism of action.

1.)  It is recommended to disclose the MK-801 in the Abstract as (NMDA) receptor blocker MK-801

2.)   Aripiprazole is known to be a partial agonist activity on D2 and 5-HT1A receptors as well as its antagonist activity at 5-HT2A receptors.It is desirable to clarify in a more direct manner why did the authors use NMDAr antagonists.  For what special purpose and why did the authors use 3 different antagonists of NMDA receptors.

3.)  It is recommended to add a sentence about the conversion of the dose from human to rat and why did the authors used 1, 3, 10, 30 mg/kg Aripiprazole

4.)  Table 2,3:  it is recommeneded  to clarify term “NMDAr antagonists’ vehicle” in a footnote.

5.)  Section 5.7: “All data sets were cleared from outliers by removing the data being beyond two standard deviations”. It should be argumented.

6.)  What a program have been used for qEEG analysis and which electrodes have been used (manufacturer). HEM software (version 4.4, Notocord France) should be referenced.

Author Response

Dear reviewer,

Thank you for your comments on our manuscript. We carefully read your different comments and edited our manuscript accordingly:

  • To better describe MK-801 as an NMDAr blocker in the abstract, we added in lines 22-23: ‘… a SZ rat model induced by the NMDAr antagonist MK-801.’
  • Schizophrenia is characterized by sustained NMDAr hypofunction that can be reproduced in animal models through genetic or pharmacological inhibition of NMDAr. This aspect is referred in the literature as the ‘NMDAr hypofunction hypothesis of schizophrenia’, which has been further highlighted in lines 59 & 60: ‘… which makes the link with SZ NMDAr hypofunction hypothesis [13] as well as in lines 80 & 81 ‘… induced through NMDAr pharmacological blockade to match the NMDAr hypofunction hypothesis’. Pharmacological inhibition of NMDAr in animal models can be achieved with different NMDAr antagonists such as PCP, MK-801 or ketamine. We therefore chose to compare these three compounds regarding their capacity to reproduce SZ features before evaluating aripiprazole, which has been added in lines 107-109: ‘We chose to test these three different NMDAr antagonists in order to compare their capacity to reproduce ASSR impairmentsand in lines 138-139: The effects of the three NMDAr antagonists were then evaluated and compared on behavioral features relevant to SZ’. Aripiprazole mainly acting on dopaminergic and serotoninergic transmission, testing its effects in a model of NMDAr hypofunction still appears relevant because NMDAr hypofunction is thought to indirectly contribute to symptoms by dysregulating the dopaminergic system, which has been further highlighted at the end of the introduction, lines 85-98: ‘Regarding behavioral manifestations specifically, NMDAr hypofunction is thought to produce further dysfunctions within mesolimbic and mesocortical dopaminergic pathways, leading to positive symptoms (e.g., psychosis and agitation), cognitive impairment, and negative symptoms such as social withdrawal in patients [25,26]. These symptoms can be studied in animal models induced by pharmacological NMDAr antagonism using various readouts such as social interaction deficit or hyperlocomotion, the latter being commonly used as a translational index of psychotic-like behavior. However, the validity of these behavioral tests has been challenged [27]. Moreover, NMDAr hypofunction could directly contribute to social symptoms by provoking an imbalance between excitatory and inhibitory inputs, resulting in glutamate spillover from synapses and dysconnectivity within and between cortical and subcortical regions regulating social behavior [19]. Considering the previous statements, blocking NMDAr in rats with NMDAr antagonists appears relevant to model SZ and offer an opportunity to test atypical antipsychotic aripiprazole’.
  • A sentence about the conversion of the dose from human to rat was added in the manuscript in lines 208-211: ‘…commonly used dose of 3 mg/kg, because these settings were previously shown to in-duce behavioral and electrophysiological effects in similar rodent models of SZ [31–33]. Of note, this dose of 3 mg/kg in rats corresponds to ~30 mg in humans, which is within the range used in clinical studies [34–37])’. Additionally, a sentence about the choice of the used doses in behavioral tests (1 to 30 mg/kg) was added in lines 263-264: ‘Doses of 1 to 30 mg/kg of aripiprazole were chosen based on previous studies reporting behavioral effects mediated by this drug at comparable doses [32,37,39]’.
  • As suggested, we have replaced the term ‘NMDAr’s antagonist’s vehicle’ by ‘Vehicle of PCP, MK-801 and ketamine (NaCl 0.9%)’ in tables 1, 2 and 3 for more clarity.
  • Lines 617-619: the sentence was modified to further describe the rationale for removing data above 2 times the standard deviation: ‘All data sets were cleared from individual outliers displaying aberrant electrophysiological or behavioral response compared to the other animals of their group by removing the data being beyond two standard deviations’.
  • The program to perform qEEG has been added in line 589: qEEG analysis was performed under Matlab (Mathworks, Inc., Natick, MA, USA).’ A description of the implants and electrodes was also added in lines 519-521: The four EEG leads (made of a nickel/cobalt-based alloy) of the telemetry implant (HD-S02 implant, Data Sciences International, St Paul, MN, USA) were fixed on the skull with screws and dental acrylic at the following stereotaxic coordinates.’

Best wishes,

The authors

Reviewer 2 Report

Comments and Suggestions for Authors

The authors report a series of studies of the effects of the atypical antipsychotic, aripiprazole, on EEG and behavioural effects produced by NMDA receptor blockade in rats. The first experiments in this series of studies provided convincing confirmation that three different NMDA antagonists produce EEG (diminished auditory-steady-state response (ASSR) and increased global gamma power) and behavioural features (hyperactivity and diminished social interaction) reported in prior studies and considered to reflect clinical features observed in patients with schizophrenia. In subsequent experiments they compared a single administration of various doses of aripiprazole with vehicle on the EEG and behavioural indices. They demonstrated that aripiprazole produced a convincing reduction in the hyperactivity produced by NMDA blockage but had no substantial effect on EEG measures. Although I am not an exert in experiments with rodents, as far as I can evaluate, the experimental procedures were well designed. In the study of social interaction, they selected a time window that avoided confounding effect of hyperactivity, In the study of hyperactivity, they demonstrated that the antipsychotic clozapine produced a similar reduction in the hyperactivity produced by NMDA blockade, as an active control. The statistical analysis was sound. The authors interpretation appears sound, apart from the authors uncritical acceptance that of the common assumption that hyperactivity in the rodent NMDA blockade model of schizophrenia is a valid model of positive psychotic symptoms. Overall, this study provides a useful confirmation of the potential utility of the rodent NMDA blockade model of schizophrenia for evaluating therapeutic effects of antipsychotics, despite the limitation imposed by use of animal models that cannot produce realistic analogues of psychotic symptoms. Furthermore, their finding regarding the lack of efficacy of aripiprazole on EEG features and on social interaction in patients reflect corresponding observations in patients. The implication in the abstract that social deficits reflect cognitive deficits characteristics of schizophrenia has limited validity and should be removed.

Author Response

Dear reviewer,

Thank you for your comments on our manuscript. We carefully read your different suggestions and edited our manuscript accordingly:

  • We modified lines 88-92 to highlight the limitations in mimicking SZ symptoms in animals ‘These symptoms can be studied in animal models induced by pharmacological NMDAr antagonism using various readouts such as social interaction deficit or hyper-locomotion, the latter being commonly used as a translational index of psychotic-like behavior. However, the validity of these behavioral tests has been challenged [27].’
  • We removed the misleading sentence that inferred that social behavior deficit reflected cognitive impairment in the abstract (line 27).

Best wishes,

The authors

Reviewer 3 Report

Comments and Suggestions for Authors

Review of the paper entitled „Differential Effects of Aripiprazole on EEG-recorded Gamma-Band Auditory Steady-State Response, Spontaneous Gamma Oscillations and Behavior in a Schizophrenia Rat Model” by Florian W. Adraoui, Kenza Hettak, Geoffrey Viardot, Magali Alix, Sabrina Guiffard, Benoît Meot, Philippe L’Hostis, Anne Maurin, Eric Delpy, Christophe Drieu La Rochelle and Kevin Carvalho

      In the presented studies, the Authors characterized an acute schizophrenia rat model induced by N-methyl-D-aspartate-receptor (NMDAr) blockade with phencyclidine (PCP), MK-801 (dizocilpine) and ketamine. The obtained results indicated that PCP, MK-801 and ketamine reduce the phase synchrony of the auditory steady state response (ASSR), increase locomotor activity and decrease social interaction in rats.

     Following these results, the Authors decided to select one NMDAr antagonist for the next set of experiments aiming at characterizing the effects of aripiprazole on the electrophysiological and behavioral features of this schizophrenia animal model. For further research, the Authors chose MK-801. The obtained results demonstrated that aripiprazole does not recover the ASSR deficit produced by MK-801 in animals. Aripiprazole also does not normalize the increase in spontaneous gamma oscillations provoked by MK-801. It has also been shown that aripiprazole does not alleviate MK-801-induced social behavior deficit.

Aripiprazole, on the other hand, reduces MK-801-induced hyperactivity in a dose-dependent manner

     This is an interesting paper.

 My comments

     The Authors should explain more convincingly why they did not test whether aripiprazole was able to correct the electrophysiological and behavioral deficits induced by all three NMDA receptor antagonists. Why was only MK-801 chosen?

     The Authors should briefly explain the similarities and differences in the mechanisms of NMDA receptor inhibition by PCP, MK-801 and ketamine [for example; Hiramatsu M, Cho AK, Nabeshima T. Comparison of the behavioral and biochemical effects of the NMDA receptor antagonists, MK-801 and phencyclidine. Eur J Pharmacol. 1989 Aug 3;166(3):359-66. doi: 10.1016/0014-2999(89)90346-4. PMID: 2553433; Zorumski CF, Izumi Y, Mennerick S. Ketamine: NMDA Receptors and Beyond. J Neurosci. 2016 Nov 2;36(44):11158-11164. doi: 10.1523/JNEUROSCI.1547-16.2016. PMID: 27807158; PMCID: PMC5148235; Lodge D, Mercier MS. Ketamine and phencyclidine: the good, the bad and the unexpected. Br J Pharmacol. 2015 Sep;172(17):4254-76. doi: 10.1111/bph.13222. Epub 2015 Jul 28. PMID: 26075331; PMCID: PMC4556466, and other ].

      Deiana et al. indicated that MK-801 impairs social recognition memory, but not sociability [Deiana S, Watanabe A, Yamasaki Y, Amada N, Kikuchi T, Stott C, Riedel G. MK-801-induced deficits in social recognition in rats: reversal by aripiprazole, but not olanzapine, risperidone, or cannabidiol. Behav Pharmacol. 2015 Dec;26(8 Spec No):748-65. doi: 10.1097/FBP.0000000000000178. PMID: 26287433].

     Sams-Dodd demonstrated that rats did not show any enduring behavioural changes as a result of the treatment with MK-801 and PCP. The author therefore concludes that although the neurodegeneration induced by NMDA antagonists has functional consequences, it may only cause cognitive impairment, but not mimic the key features of the schizophrenic symptomatology such as positive and negative symptoms [Sams-Dodd F. (+) MK-801 and phencyclidine induced neurotoxicity do not cause enduring behaviours resembling the positive and negative symptoms of schizophrenia in the rat. Basic Clin Pharmacol Toxicol. 2004 Nov;95(5):241-6. doi: 10.1111/j.1742-7843.2004.pto950507.x. PMID: 15546479].

I am asking the Authors to comment on the results cited above in the Discussion.

Author Response

Dear reviewer,

Thank you for your comments on our manuscript. We carefully read your different suggestions and edited our manuscript accordingly:

  • Our choice to test aripiprazole’s effects only on the model induced by MK-801 was further described lines 195-203: ‘The choice of continuing with only one NMDAr antagonist was first motivated by ethical reasons, as evaluating aripiprazole using a single NMDAr antagonist allowed to reduce the number of animals used. However, investigating the effects of aripiprazole on all NMDAr antagonists remain of interest, as this could lead to valuable scientific knowledge. Thus, our choice was to continue with MK-801 due to its better potency towards NMDAr and wider commercial availability compared to the two other drugs. Moreover, the effect of MK-801 on locomotor activity has been shown in our study to not overlap with its effects on social interaction, which has not been investigated for PCP and ketamine.’
  • To further discuss the different effects of the three NMDAr antagonists (PCP, MK-801 and ketamine), we added a small paragraph (lines 322-331) in the discussion integrating references suggested by the referee: ‘Although the reductions in the ASSR PLF and social behavior were similar for the three NMDAr antagonists in our study and that these molecules have a lot of charac-teristics in common, it is still worth noting that they also display slight differences. For instance, ketamine is less potent towards NMDAr compared to PCP and MK-801, and these three NMDAr antagonists also display different affinities towards each subunit of the NMDAr [Zorumski et al., 2016; Lodge et al., 2015]. These aspects might for instance explain why previous studies found that different neurochemical and behavioral profiles could be induced by these three compounds [Deiana et al., 2015; Zorumski et al., 2016; Hiramatsu et al., 1989], and why ketamine produced a weaker hyperlocomotion and was used at a higher dose compared to the two other compounds in our study. ‘. We did not include the recommended reference Sams-Dodd et al. 2004 as they used a sub-chronic administration of MK-801, which is out of the scope of our current study.

Best wishes,

The authors